# Engagement to Enhance Community: An Example of Extension's Land-Grant Mission in Action

**Cheryl Burkhart-Kriesel** [1,*]**, Jason L. Weigle** [2] **and Jennifer Hawkins** [3]

[1]   Panhandle Research and Extension Center/Department of Agricultural Economics, University of Nebraska-Lincoln Extension, Scottsbluff, NE 69361, USA

[2]   Southeast Research and Extension Center, University of Nebraska-Lincoln Extension, Hebron, NE 68370, USA; jason.weigle@unl.edu

[3]   Extension Regional Center, University of Minnesota Extension, Rochester, MN 55904, USA; hawki044@umn.edu

*   Correspondence: cburkhartkriesel1@unl.edu; Tel.: +1-308-632-1234

**Abstract:** Engagement is a foundational practice for the Extension systems of land-grant universities and is demonstrated through its' work in partnership with individuals, organizations and communities. This article will share how an Extension-led effort, focused on an aspect of community development, integrated several components of engagement starting with the initial conversation through the evaluation process. Practitioner reflections on two examples that occurred in different states will highlight the processes and tools that helped nurture engagement between faculty and community and support the development of a sustainable and resilient community. The multi-state implementation will illustrate the unique depth and breadth of public participation that can be achieved when academic institutions are focused on engagement to strengthen communities.

**Keywords:** community engagement; civic engagement; participation; involvement; community development; community vitality

## 1. Introduction

Community engagement may seem like a new function for institutions of higher education but it is not new for land-grant institutions. Engagement, or civic participation, is embedded within the mission of colleges and universities that collectively represent the land grant system. The system, as we know it today, was established through a series of Congressional actions. In 1862, Congress passed the Morrill Act that allowed federal land to establish education institutions in each state. Several years later in 1890, a second Morrill Act established educational institutions for black students. In 1994 tribal colleges and universities were officially added to the land grant system. Initially, land-grant institutions focused solely on teaching degree-seeking students. The Hatch Act of 1887 added research as a mission. Finally, in 1914, the Smith-Lever Act added outreach and extension. The 1914 Act led to the creation of extension services and articulated: "The third mission of the Cooperative Extension Service challenged this unique set of colleges to extend their resources to solve public needs through non-formal, non-credit educational programs." (Muske et al. 2007, p. 4).

Extension professionals have historically relied on timely, research-based content and interpersonal and group process skills to make a connection with the people they serve. Their knowledge base has mirrored the evolving needs of society, from the initial adoption of innovative farming practices to today's inclusion of STEM (science, technology, engineering and math) in youth-based opportunities (Gould et al. 2014). Interpersonal and group process skills linked to engagement have also evolved and broadened from early field demonstration projects with Extension educators to today's web-based sessions and applications using real-time virtual interaction.

The style of engagement has also evolved over time. Throughout its history, there has been an underlying tension between the role of the Extension professional and the learner (Peters 2002, 2014). Extension has used variations of the expert delivered (one-way) model, the reciprocal engagement (two-way) model as well as applications that incorporate aspects of both models. Recent research indicates that this hybrid model, which utilizes data and expertise but also incorporates the tacit knowledge of the learners, is becoming more common. This co-learning approach blends both approaches, leveraging the benefits of each (Vines 2018).

Extension, by nature, lives at the intersection of research and practice. Extension programs emerge from academic research endeavours. Programs are applied, tested and refined with stakeholder input and through the self-reflection of Extension educators. The educators then evaluate and disseminate their findings and experiences for both academic and practitioner audiences.

Extension's direct and indirect involvement with community development has evolved over time and in many ways mirrors the engaged practice of community development. In order to understand this evolution, it is helpful to take a closer look at how both engagement and community development are defined and function.

In this article, the authors explore engagement for the purpose of community development through the delivery of a program called Marketing Hometown America (MHA). MHA is a facilitated process through which communities explore their assets and how these assets can be utilized to retain and attract new residents. The engagement fostered by MHA not only helps them to identify and celebrate community assets, it brings community members together in collective action. This collective action, in turn, builds capacity for potential future action and collaboration as new situations arise.

We examine the outcomes of MHA through the lenses of community, engagement for community well-being and how engagement endeavours of University academics inform and influence community development. Specifically, this article focuses on how engagement can foster social interaction, break down barriers and help residents recognize aspects of their community which others may find attractive. We utilize our personal and group reflections on the process and the experiences of two purposefully selected communities who participated in the program to explore implications for engagement. Our conclusions provide insight for University faculty and educators can use the findings to better interact with communities to strengthen resiliency.

### 1.1. Community Defined

Community, as a term, has a wide variety of uses and connotations in both colloquial and academic contexts. In the early history of the United States, one's interactions were limited to a specific geography where most needs were met (Kaufman 1959). As the nation's economy became more industrialized and urbanized, places of residence and production were no longer one and the same. The disconnection of production and residence extended linkages beyond the place where one resides, increasing the dynamics of the community concept (ibid).

The expanded sphere of interactions and the increasing importance of those interactions in understanding community spurred the development of many different approaches to defining 'community.' Hillery's (1955) seminal piece sampled the literature of the time, collating commonalities among definitions of community. His research showed 70 percent of definitions cited an area, common ties and social interactions as important features of community.

In summarizing community and community development, Theodori identified two applications of the term 'community' in academic literature: territory-free and territory-based (Theodori 2005, p. 662). Territory-free communities generally "describe types of social groupings or networks" such as "'the business community,' 'the farm community,' and/or 'the Hispanic community,'" or are driven by the expansion of the internet and early forms of social media. The latter term, place-based, is based on "geographically localized settlements" or "territory-based communities."

Community, in the context presented in this paper, refers to a place-based group. As will be described, the emphasis of the Extension program is engaging residents in developing place-centred

actions that improve the attractiveness of said place to new residents. Accordingly, a place-based approach to defining community is warranted.

## 1.2. Interactional Community Theory

One area of particular interest to this endeavour is interactional community theory. This perspective, based on the psychological field works of Lewin (2000) and Mey (1972) and further refined by Kaufman (1959, 1985) and Wilkinson (1970, 1972, 1999), examines the relationships of residents as they come together to pursue place-oriented action.

An interactional field is defined as an "organization of actions carried on by persons working through various associations or groups" (Kaufman 1959, pp. 10–11). Social fields are a type of interactional field, which are the "sequence of actions over time carried on by actors generally working through various associations" (Theodori 2005, p. 663, emphasis in original; see also Wilkinson 1999). Social fields are groupings such as faith-based organizations, government agencies, environmental groups, economic development groups and so forth. (Theodori 2005).

When social fields overlap, the community field can emerge (Wilkinson 1999; Theodori 2005). As Wilkinson states, "[a] community field is a process of interrelated actions through which residents express their common interest in the local society" (Wilkinson 1999, p. 2). Social interaction, according to Wilkinson, "delineates a territory as the community locale; it provides the associations that comprise society; and it is the source of community identity" (Wilkinson 1999, p. 11).

What separates a community field from social fields is "the generalization of locality-oriented actions across interest lines" (Theodori 2005, p. 665). Generalization connects various social fields together, providing a base for community development to occur by pooling interest, expertise and resources (Theodori 2005) and community response and adaptation to threats (Flint and Luloff 2005, 2007; Weigle 2010). Thus, community emerges from the collective action of individuals, acting through the variety of groups and organizations they belong to, which express the shared interests of the generalized social field toward their shared locality.

## 1.3. Engagement as a Means to Bring Forth Community

Community, from the interactional perspective, is not inherent—it emerges from the actions and activities of those living within an area. Barriers to people assembling in social fields, being able to express their thoughts and ideas and being involved in decisions made that affect them are all challenges to the emergence of community.

A number of engagement strategies are available to address the barriers within an area as well as give residents a voice in making decisions. The Spectrum of Public Participation (IAP2 2018) developed by the International Association for Public Participation (IAP2) is one such tool. The spectrum describes a variety of approaches for including the public in decision making. It can be conceptualized as a graph with goals for engagement on the vertical axis and levels of increasing impact on decision making on the horizontal (Figure 1).

At the lowest end of the IAP2 spectrum is inform, which generally means people are informed of decisions and information and not fully included in decision making and power structures. From there, participation grows from consult (obtaining feedback from public) to involve (engaging the public to define solutions and aspirations) to collaborate (public engagement and decision making is incorporated into the process as much as possible) to empower (the public makes the final decision).

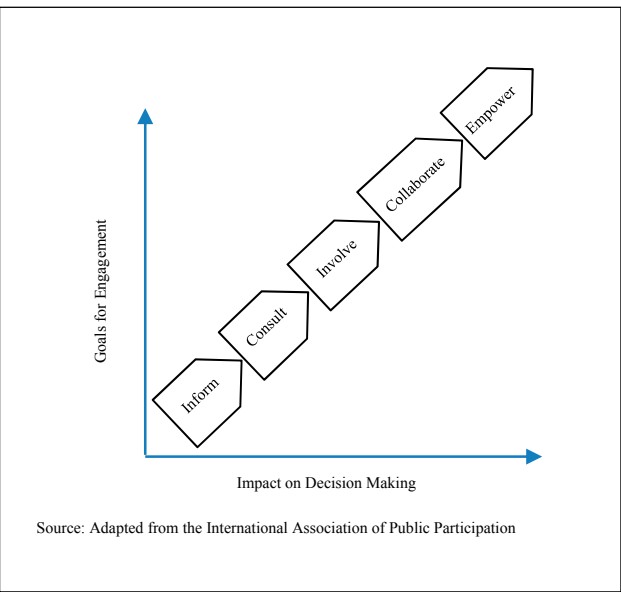

**Figure 1.** IAP2 Spectrum.

Chaskin (2001) suggests another approach. Chaskin's model articulates six dimensions that highlight the interconnected nature of community capacity, the need to build connections across different units and levels within a locality and the necessity to address barriers preventing effective creation and implementation of community capacity (Figure 2). The first dimension defines four characteristics of community capacity. The second dimension assesses the levels at which community capacity is built (individual, organizations, networks). The third assesses the function of community capacity, or, as Chaskin describes, 'the intent of engaging particular capacities (dimension 1) through particular levels of social interaction (dimension 2) to perform specialized functions (dimension 3) such as building a local capacity for planning and governance for the production of a particular good and service . . . or for informing, organizing and mobilizing residents toward collective action (299).' The fourth dimension assesses strategies for building community capacity, or the means through which the first three dimensions build toward enhancing local capacity. The fifth looks at the 'mediating circumstances' (299) which help or hinder community capacity building. Last, the sixth dimension describes the outcomes of the community capacity as it may already exist in a locality or which can be built through a community capacity building process.

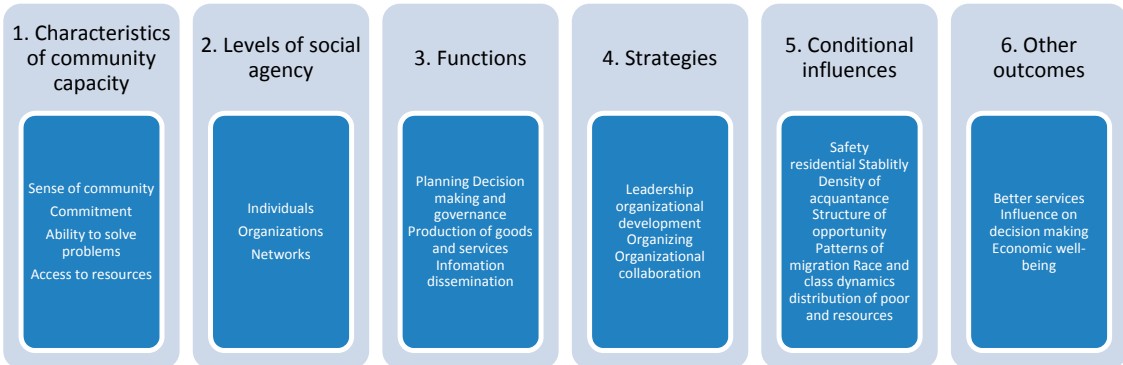

**Figure 2.** Chaskin's Model.

Participatory action research (PAR) is another way to overcome barriers to effective community engagement. PAR includes participants in the process of designing and implementing the study (McIntyre 2008; Wimpenny 2010; see also Baum et al. 2006 for a concise glossary of PAR concepts and

terminology). In this method, residents and researchers engage in defining the research and process together. It is action-oriented and based on the lived experience in a particular locality. At the same time, it is grounded in research and oriented toward the creation and growth of new knowledge.

PAR, like interactional community and Chaskin's community capacity model, is built upon the foundation of the shared experience of people as they live and interact within a place. PAR is particularly high in the IAP2 spectrum because of the integration of study participants in the creation and implementation process. With these concepts defined, we now turn to discussing the impact of these approaches on the people involved.

*1.4. Community Engagement for Well-Being*

Well-being is an important measure of how various forces, both material and non-material, interact with and affect, quality of life. Wilkinson (1999) suggests well-being consists of three components: personal, ecological and social. As they interact, the foundation of community well-being is established.

Personal well-being is affected by a wide variety of social, psychological and personal experience factors and differs greatly from person to person. Perhaps the most widely known model of personal well-being is Abraham Maslow's five levels of need: physiological, safety, belongingness/love, esteem and self-actualization (Maslow 1943). While others have added to this, Maslow's Hierarchy of Needs outlines the basic things we all need to be healthy and happy.

Ecological well-being reflects the health of the environment in and around the community and is central to both social and personal well-being because it is the source of the materials needed to fulfil lower-order needs. Ecological well-being all too often comes at the expense of personal and social well-being, as the latter two are often tied to economic well-being that drives many common quality of life definitions (Dasgupta 2001).

Social well-being is the measure of the quality of a person's interactions in a community (Wilkinson 1999). This measure includes the support systems in place to promote and sustain personal well-being, the opportunities available to take advantage of them and their impact on the community and person.

Wilkinson lists five important items that a community needs to support social well-being: (1) distributive justice; (2) open communication; (3) tolerance; (4) collective action; and (5) communion. Distributive justice is the belief that all people are equal and removal of barriers and inequalities increase communication and interactions within a community. Open communication refers to the efficiency, effectiveness and integrity of communications among people; any barriers in communication are barriers to social well-being. Tolerance refers to the acceptance of the values and beliefs of others and is also an integral component of personal well-being. Collective action refers to the degree to which people in a community work together. Communion is the celebration of community and the relationships that exist within it.

When looking at the three components of well-being—personal, environmental and social—together, we can see their importance for the formation of social fields and community fields. Well-being is about one's ability to engage with and participate in, activities promoting self- and community-health. It is about creating places and environments that provide necessities and promote interaction and engagement amongst community members. Well-being is also about sharing a combined understanding and sense of place. This is important when considering how to design and implement models to engage community capacity.

Two common models for community development are the Community Capitals framework and Asset Based Community Development. For instance, when utilizing the community capitals framework, it is critical to understand the barriers to integration and communication when looking at bridging and bonding social capital within and across the place being studied (Green and Haines 2008). Asset based community development, or ABCD, (Kretzmann and McKnight 1993) is another approach. It can be informed by well-being, especially as related to access to various forms of capital, ability to 'see' available assets and in defining what is needed in order for an effort to succeed.

The implications for community work are apparent. Anything acting as a barrier to well-being is also a barrier to social and community field formation because it limits the ability of individuals to engage in activities that create social fields. When social fields cannot form, community fields are also impeded, which means that community is unable to emerge. When social and community fields fail to form, community members are unable to leverage the assets and capitals effectively. Community development programs may fail to create sustainable change if they do not take into consideration the wide variety of factors that influence well-being and social field emergence. The literature reminds us of the imperative to address the barriers to communication, tolerance, communion and distributive justice that can prevent collective action—the foundation of community.

*1.5. The Role of University Extension in Community Engagement*

The land-grant university based Extension system has an extensive history of engagement with residents (Peters 2002; Peters 2014; Vines 2018). Extension educators have long been integrated with and into the places they serve. Since its inception, Extension's approach has incorporated elements of both the expert-driven and reciprocal engagement models. Some feel the reciprocal engagement model, which brings stakeholders into the process as co-learners, is gaining more prominence (Vines 2018).

Efforts, such as the eXtension Impact Collaborative, are also expanding the concept of innovation and community engagement in program planning and design (Vines and Stiegler 2018; eXtension Foundation 2018). Others, such as Mark Lubell, have used an expanded version of engagement and dialogue to express this new model as Extension 3.0 (Lubell 2018). It is within this nexus of education and engagement that Marketing Hometown America (MHA) exists. It could be seen as a hybrid approach (Vines 2018) that emerged from new knowledge that was translated by Extension team members into an initial program of actionable activities. The Extension team member engages in a reciprocal implementation with the community. Through the process, the community better understands its assets and gaps and is able to articulate a plan detailing how residents can work together to enhance community, attracting and retaining residents for sustainability and resiliency. At the same time, the experiences of the community inform Extension professionals about the practical implications of the program and process. These insights, coupled with ongoing academic research, allow for continuous refinement of the methodologies by which future communities participating in the program might pursue their goals.

## 2. Materials and Methods

Reflective practice is the primary method used to obtain a level of understanding about the types of engagement practices implemented and the effectiveness of those practices in this study. The use of self-reflection has a long history, as noted by John Dewey, American educational reformer: "We do not learn from experience, we learn from reflecting on experience" (Dewey 1933, p. 78). Biographical research is a unique and grounded method that can provide valuable insights into professional practice (Dausien et al. 2008). As a common educational learning strategy, self-reflection is often integrated into aspects of health care (Reflective Practice 2018) which focuses on client care and business which is often linked to employee productivity (Stefano et al. 2014).

Reflection can be established as both a learned and practiced habit by professionals. Typically as individuals reflect back of the situation they ask basic questions of themselves about the experiences they have encountered such as, "What happened? Why did it happen? What was the situation or context? What was my role? What did I learn and how can I use this in the future?" The positive impact from reflection is not only connected to task understanding but it also is linked to self-efficacy (Stefano et al. 2014).

Reflective practice can also be considered an active learning strategy (Meyers and Jones 1993). It is assumed that a reflective professional will: (1) acknowledge all practice as a learning opportunity; (2) think about what you do and the meaning of your practice experience regularly; (3) create opportunities to share your experiences with others; (4) examine the assumptions behind your practice; (5) compare

the theory of your practice to what you actually do; and (6) consider the systemic influences that impact your practice, imagine positive improvements and advocate for these through action (Kinsella 2001, as adapted by Reflective Practice 2018, p. 3).

In this study, two of the three authors of the manuscript were field practitioners who had the opportunity to not only document individual reflections but they also participated in monthly multi-state video conference meetings that allowed for a sharing of peer reflections and insights. The third author provided an outsider's perspective of what he saw in the data and gave insights at a more holistic and overarching level. The two program examples, one in Minnesota and the other in Nebraska, were created from data made available through journal notes, interviews with faculty directly involved in the program implementation and through peer-to-peer conversations that were documented in the monthly video conference meeting notes. The informal nature of the group video conference conversations allowed for issues to emerge during the program process (reflection-in-action) and after the program had been completed (reflection-on-action) (Schon 2009).

The primary material needed for reflection is the allocation of time to critically assess what happened and what was learned. Journaling is often cited as a tool to help spark regular and deliberate reflection (Stefano et al. 2014). In this study, 10 peer-to-peer video conversations that happened over the course of 18 months were the primary method used to reflect on practice with journaling being a secondary method. As could be expected, the quantity and quality of the journal notes varied greatly across the 25 team members. The two locations were chosen, in part, because of the detailed journal notes and/or the in-depth conversations that brought to life the engagement opportunities available within the locations. An additional tool was the documentation of events coupled with the insights and shared perspectives of all the practitioners on the team as the events unfolded. An example of an event is the formation and implementation of community action teams following the program Action Forum. Insights on these events were documented through the video conference meeting notes.

The focused analysis of engagement within the program was elevated through the development of this manuscript. The literature review allowed for conversations into the theoretical underpinnings grounding community formation including interactional community theory, engagement for community well-being and how engagement endeavours of University academics inform and influence community development. The in-depth conversation between members of the writing team using video conference meeting and journal notes was an opportunity for reflection and consensus by comparing and contrasting different program locations as well as exploring broader, or macro level, engagement.

## 3. Program Summary

The authors feel that an illustration of an engaged model of University supported community development is found in the MHA example. It supports community resiliency and sustainability by engaging citizens in a focused process for considering resident recruitment factors and tactics.

### 3.1. Program Context and Methods

Between 2007 and 2012, Extension educators in Nebraska, North Dakota and South Dakota teamed up to pursue research exploring rural community resident recruitment and retention (Cantrell et al. 2008; Burkhart-Kriesel et al. 2014). Two research projects were designed around this issue and garnered funding from the U.S. Department of Agriculture. Through both survey research and focus group interviews, new residents in those three states told researchers that rural communities did not promote or showcase their communities in a way that matched the needs of today's potential new residents (Burkhart-Kriesel et al. 2014). The findings from this research sparked the creation of a community engagement program called Marketing Hometown America in 2013 with a grant from the University of Nebraska Rural Futures Institute.

The goal of the program is to improve new resident recruitment and retention. Program designers anticipate two outcomes. First, it is designed to help communities discover and take action, on

opportunities to enhance their town as a place to live and work. Second, through the process communities uncover what makes them unique, who might be looking for those unique attributes and how to market and promote those assets to those who would be interested. As community members participate in the program they: (1) learn what new residents are looking for as they relocate to a rural community; (2) discover often overlooked local assets that attract potential new residents; (3) use positive conversations to begin or expand community marketing; (4) create a welcoming culture needed to attract new residents; and (5) build and implement a marketing action plan.

The program follows the study circle process developed by the national organization *Everyday Democracy* (Figure 3). After a community coalition or sponsoring group is organized the group sets location specific goals and a plan of action. Local facilitators are trained by Extension professionals on the program content and group process, thus building long-term skills and local capacity. After the facilitators are trained, participants are recruited to join the study circles. An initial community event is held to increase program visibility and to offer one final opportunity for participant recruitment. Then small dialogue groups or study circles are formed and led by these trained local facilitators.

These circles of between 8–10 people meet four times, for about two and a half hours each time, using a guidebook that directs participants through a series of discussion questions and activities. After the fourth group session, the study circle participants along with members of the community participate in an Action Forum. At the Forum, the circles share their plans, using a template and attendees can vote on their favourite parts of the plans and activities presented, regardless of whether or not they have been a part of a formal study circle group. The plans highlight: (1) new resident research (who is coming here now and why; what skills/expertise does our community need); (2) target markets (what groups match up with our community assets); (3) a potential message and reach of the market (what community benefits or value is offered to the identified group; what is the overall message; what techniques do we use to reach these groups); and (4) additional actions the community can do to make the community more marketable (what do we need to improve and what actions can we do to make this happen). The highlights of the plans are typically listed on flipchart paper and posted around the room for everyone to view following each presentation. Behind the scenes and prior to the Forum, the study circle facilitators meet to develop a summary chart of all of the plan highlights so that any duplication of ideas can be consolidated. This summary flip chart paper listing is placed in an obvious location in the room so that Forum participants can vote on their favourite aspect of the plan by placing their allocated sticky dots on the selections they feel are a priority for action. Community members then volunteer to either lead or help with one of the selected priority actions that emerged from the voting process. Extension professionals serve as a guide for the community throughout the entire process. Figure 3 illustrates the process. For more information go to Supplementary Materials: https://youtu.be/Si75I5qgd6I, YouTube video.

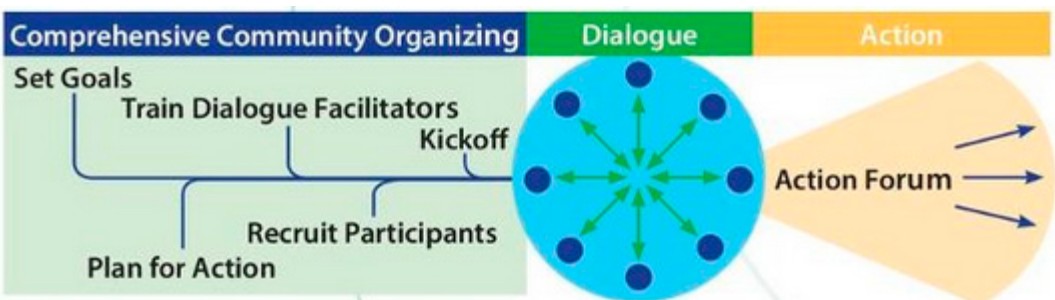

**Figure 3.** Study Circle Process. Source: www.everyday-democracy.org (A Guide for Training Public Dialogue Facilitators, 37).

Marketing Hometown America was originally offered in North Dakota, South Dakota and Nebraska. Interest grew and currently Minnesota and Iowa offer the program as well. To highlight

unique aspects of engagement, two locations, one in Minnesota and the other in Nebraska, are featured below.

### 3.2. Minnesota—City of Litchfield

Located 90 min west of Minneapolis, Litchfield, is a town of about 7000 people. The community credits a talk given by a University Extension researcher on the "brain gain" effect, which refers to an increase in the 30–49 year old population categories in rural areas, as the spark for community action on how to attract and retain residents. In the words of Chamber Director David Krueger, interviewed for episode two of an Extension produced podcast (Kallevig 2017), "It really was Ben who sort of opened the eyes of a lot of people that rural is not as desperate as people seemed to think and the narrative out there that rural has been shrinking and diminishing, when really the population overall has remained very close to the same. In many cases, including Litchfield, lots of jobs have been retained and modernized and overall the income is higher in Litchfield, as well as Meeker County." He continues, "I think that after hearing some of this, rewriting that picture that we're not as desperate as some people would like rural to be. We wanted to take a look at how we really market our community, to get the word out there that we have a lot to offer and we want to compete and we want to be out there to toot our horn about what's going on in Meeker County as well as Litchfield."

The community leaders also credit their County Extension Committee, a local advisory body that supports Extension's local connections with stimulating their participation. Krueger noted, "None of this could really be possible without the Meeker County Extension Committee; they are the ones that really helped fund some of this and encouraged us to take a look at this. I did not want to leave them out of that conversation."

With this mindset and source of support from the County Committee, community leaders engaged the local Extension educator to explore what they could do. They decided to pursue the Marketing Hometown America program, which in Minnesota is branded as "Making it Home."

The application of the program followed the basic model for other installations of the program. The Extension educator served as a guide, supporting the implementation process, sharing demographic insights and training facilitators. Community leaders, including those from the Chamber of Commerce and city's economic development office, convened a steering team of six. From there, they engaged fifteen community members to serve as facilitators. The process involved 130 community members.

A unique element for Litchfield was their participation in a cohort-based leadership training offered by a Minnesota foundation. The leaders of the group believe that training provided some basic tools that this program was able to leverage, including social capital among group members, a common language about community development and what supports it and energy to put those ideals into action. In the words of Kruger: "Having that social capital infrastructure is really important, because you can start a program but if you do not have enough people to facilitate and you do not have enough actionable people that can help move the findings later, that it stalls out." He continues, "So you really really need to make sure that social capital infrastructure is there, or you build it before you end up in a program like this."

The action forum brought together more than 70 community members to review the discussion results and prioritize action items. Eleven specific activities were identified and teams formed to move those ideas forward. Ideas ranged from communications (developing a website, sharing positive stories) to physical elements (a community centre, dog park). The leaders of the local effort reiterated the value of the action oriented approach of the Marketing Hometown America model.

### 3.3. Nebraska—York County

York County is home to about 14,000 residents and located approximately 40 miles west of Lincoln Nebraska. Its largest community, York, has a population of 8000 and serves as the county seat. There are several smaller towns in the County with populations of under 1000 each. With the

Interstate—80 system dissecting the county east to west, it has historically taken advantage of its location as an agri-business and transportation hub in the region.

Both a housing and workforce development study, completed earlier and initiated by York County Economic Development, set the stage for a more focused conversation on the larger issue of new resident recruitment and retention.

The program followed the basic engagement process outlined in the Marketing Hometown America study guide with 150 people participating in the 20 study circles that were formed. What made this location unique was the timing, the county-wide approach and the leveraging of key assets within the community.

Timing of the intervention was important. The Economic Developer identified MHA as a process that would complement two studies focused on local housing and workforce development. The steering committee was well-networked with well-defined goals. The group who provided guidance to the program had broad community representation—it was both well connected and functioned at a high level. The local representatives from the college, hospital, government and economic development were all engaged and led study circles.

Adjustments were made to accommodate the county-wide approach. With a significant portion of the residents involved in agricultural production, the steering committee planned the program elements to coincide with the winter months, allowing agricultural producers and agri-business owners to be more engaged. It also allowed the group some additional time to reach out to organizations in the smaller communities across the county to create a "buzz" around the topic of new resident recruitment. The Extension faculty started the conversation with the Economic Developer in the late spring with the program launching in the early months of the following year. A countywide email listserv that included 90 people also helped with the communication process. To gather additional input, over 300 people participated in the community report card related to the MHA process.

The Action Forum resulted in six major themes or action groups being formed. They included: (1) promotion of community strengths especially quality of life; (2) create a welcoming atmosphere (possible targets were alumni groups and tourists); (3) develop a comprehensive "story telling" effort to include an app, website and task force to develop the plan; (4) enhance marketability through improved housing opportunities; family support (especially child and adult day care opportunities) and the promotion of being "open for business" as a way to encourage potential new residents to see untapped income and employment opportunities.

Five smaller communities plus the town of York were involved in the process. One of the communities, Bradshaw, with a population of 272, was featured in the county newspaper two years after the program was implemented for their continuing community work as a result of the program. One of the goals Bradshaw had identified was to connect the local youth in the community. While there is not an elementary school, there are still many youth in community. Activities since the program ended included refurbishing the basketball court, extending community centre hours and improving community celebrations to include more youth related events. A local volunteer recognized that the process helped them to understand their strengths, "There are so many great things in this town. Some of us have lived here so long we look past them. There are more good things around us than some of us realized" (Votipka 2018).

## 4. Discussion

The authors acknowledge that the data from two program sites is insufficient to make broad sweeping generalizations about engagement within the Land-Grant University Extension system. However, we feel there are several "lessons learned" from the reflective process that the multi-state program team adopted. These lessons, or insights, may assist faculty contemplating involvement in engagement models to enrich their academic experience.

Place based action through an engaged process can lead to an empowered community. Thinking back to our earlier review of the literature related to the program, the MHA effort is an attempt to

engage a place-based community to come together around place-oriented action. The use of the study circle methodology brought Wilkinson's interactional community theory to life. The community members engaged with Extension and the MHA program to pool interest, expertise and resources to define a community response to a threat. In this case, the threat is population loss or missed opportunities to attract and retain residents. The creation of action plans and formulation of teams and tactics to implement these plans exemplifies the emergence of community.

Within the Marketing Hometown America example, intentional engagement and public participation is embedded within the community work undertaken by Extension professionals in support of sustainability and resiliency. Engagement is a foundational element in this program, not an "add on" or additional step. The work of the academic Extension professional exemplified all elements of the IAP2 spectrum, beginning with the act of informing about a problem and a process to address the problem. The work continued along the spectrum engaging a study circle process allowing citizens to opportunities to interact with the process from consultation through empowerment. Community members had the opportunity to:

- Learn about the program through media (social media, radio, newspaper) or word-of-mouth (Inform)
- Participate in the kick-off event (Inform)
- Participate in informal surveys on what current and new residents want and need (Consult)
- Join a steering committee or organizing group (Involve)
- Join a study circle (Involve)
- Serve as a study circle facilitator (Collaborate)
- Participate in the action forum (Empower)
- Volunteer for an action team (Empower)

MHA incorporated various tools to encourage participation and engagement. The full study circle process is designed to include community coalition building, community events and individual engagement through discussion groups and action teams. All of these engagement opportunities rely on civic participation. Additionally, MHA takes an asset based approach to community development. The study circle participants seek to understand community assets that make it special and unique and how they might leverage those assets to attract and retain residents.

In addition to the program process, an evaluation technique, Ripple Effect Mapping, was used with some of the communities to understand the intended and unintended outcomes and impacts from the program. The technique is applied six to 12 months following the program and invites 8–12 people to come together to reflect on the outcomes of the program. The facilitator of the process asks the participants to reflect on what happened as a result of the program starting with the obvious outcomes and keeps asking, "What else happened?" to start building additional outcomes or "ripples" from the initial actions. The process reveals new discoveries where actions or discussion influenced and spun off additional activities or outcomes.

This technique is helpful in capturing both the intended and unintended outcomes (Chazdon et al. 2017). The community can then see what happened through this process and it gives them and the evaluator, additional documentation about the scope of the actual changes resulting from the program. Pivotal to Ripple Effect Mapping is the engagement of the community in pinpointing the various outcomes and linkages from their perspective.

The Ripple Effect Mapping technique was used as a tool to ascertain the impacts of the MHA program. Through the exercise, participants documented intended changes occurred, such as marketing actions and amenity improvements. They also documented unintended changes including an increase in engagement of both adults and youth in the community, expanded leadership, increased connections and networking and enhanced awareness of civic engagement opportunities and community pride. These responses provide evidence that the program is enhancing well-being through community engagement.

Although it may be too early to indicate if the community field has fully emerged as a result of the program, one bit of evidence shows social fields coming together to address the common good. In the community of Bradshaw, a part of the York county program in Nebraska, a 2018 news article documented community work credited to the MHA program two years after program implementation. The resident's acknowledgement of a new collective enthusiasm and set of actions is one signal that the community field has or is emerging (Votipka 2018).

Finally, the program is an illustrative example of the hybrid model of University-community collaboration. The program emerged from research and interaction with the community. As it is implemented within communities across several states, lessons and insights emerge. Extension professionals engaged with the program are using these insights to further refine community development efforts and craft meaningful research agendas to further new knowledge and application.

Beyond these insights, the authors offer several themes, which emerged from practitioner reflections and may inform future efforts by academic professionals to engage communities in a reciprocal model. These insights are described below:

1. Engagement takes time.

Working with groups and coming to agreement on next steps and actions can be time consuming. Each community has its own rhythm. A University timeline cannot supersede the community timeline if an effort is to be successful. It also takes time to overcome barriers within communities. If not overcome, these barriers can prevent meaningful access to the process for community members.

2. Flexibility is key to accommodate participation.

A fixed approach to engagement presents challenges in the context of community. While the framework for the program was set, modifications were made to accommodate maximum citizen participation. For example, in York County, multiple training sessions were provided in an effort to accommodate differing schedules. Building adaptability into engagement programs, through an iterative process such as Marketing Hometown America or the eXtension Innovation Kit process (eXtension Foundation 2018), can help educators and practitioners turn barriers into opportunities.

3. Timing influences the process.

In both locations included in this analysis, previous efforts to build social capital catapulted this program forward. The existence of these social networks allowed for community members to more easily mobilize around this particular effort and connect community resources. Having these networks and relationships established prior to engaging with the MHA model allowed for deeper, more honest conversations and work on new resident recruitment.

4. Getting the right group around the table as a steering committee can make communication and action easier to accomplish.

Similar to the prior point, past positive history in working together to get things done makes it much easier to take on new issues and efforts. While these smaller victories are a beneficial starting point, practitioners still need to ensure that the kick-off groups are representative of all facets and not just a continuation of existing power structures.

5. Design the process to make room for multiple perspectives.

When discussing how to recruit new people to an area, a default for communities and community members is to approach the questions through the lenses of their personal experience. Thinking back to interactional community theory, a process designed to ensure multiple, diverse perspectives are involved builds community, contributing to greater resiliency.

6. Celebrating progress enhances meaningful change.

The Ripple Effect Mapping method revealed the value of acknowledging progress and success resulting from the activities. Celebrating can create a flywheel effect, propelling the effort forward indefinitely.

## 5. Conclusions

As with any effort, there are lessons to be learned from these examples of community engagement. Following the reflective practice, faculty engaged with delivering Marketing Hometown America found value as they thought back on their experience in an effort to better understand the engagement process.

Marketing Hometown America is an example of how educators can engage proactively and positively with residents to help form community. The interactions generated by the program create social fields and these social fields can, in the right circumstances, form community.

The community field is critical as it forms the foundation of interactional capacity, or the ability to leverage networks and resident capacity to adapt in a time of need (Flint and Luloff 2007). Interactional capacity, combined with breaking down barriers to personal, social and ecological well-being, can help communities respond and adapt to change without losing their identity and sense of place, a critical component to long-term community resilience (Weigle 2010).

As others have noted, there are a wide variety of measures used to document community resilience to change (Sherrieb et al. 2010). In this paper we have focused on how engagement can help foster social interaction, break down barriers and help residents recognize the aspects of their community which others may find attractive. While the outcome of the program is important, what is perhaps more important is the long-lasting relationships, partnerships and networks that are formed by bringing people together. By integrating engagement into research and practice, we can lay the foundation for community resilience in the face of ever-constant change.

**Supplementary Materials:** The following are available online: (1) https://youtu.be/Si75I5qgd6I, YouTube video that highlights the community engagement process and stakeholder comments; and (2) http://viewer.zmags.com/publication/29de58a3; an e-magazine modelled after the program study guide (provides an overview of the program process and materials available).

**Author Contributions:** Conceptualization, C.B.-K., J.L.W. and J.H. data curation, C.B.-K. and J.H. writing–original draft preparation, C.B.-K., J.L.W. and J.H.

**Funding:** This research received no external funding.

**Conflicts of Interest:** The authors declare no conflict of interest.

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
