# Peer review of "Engagement to Enhance Community: An Example of Extension’s Land-Grant Mission in Action"

_socsci, doi:10.3390/socsci8010027_

Round 1

Reviewer 1 Report

This is a well-written article that offers some valuable points and contributions to the community development literature, with a specific focus on the Extension system in the U.S.  I do have a few items to share with the authors that I hope will further strengthen a paper that is already in decent shape. 

Lines 41-42: I'm not sure I would agree that it's passive learners when it comes to the expert model.  It is more the fact the learners may not be major contributors when it comes to the content, but they are, in some cases, very active learners (such as field days). I would recommend this be modified just a bit. 

Line 43: The notion that the collaborative model is a more recent phenomenon needs to be clarified.  What is recent in terms of the time period?  I believe the CD profession (see the CD Society principles of good practice) has some some time advocated collaboration in community work. So, "recent" does not offer much clarity as to when this happened.  I also would recommend you add citations to support this statement. 

Line 47:  I would suggest that Extension programs are not solely the result of research conducted by the LGUs.  There are several other universities and research centers that have generated knowledge that is being incorporated into the work of the LGUs outreach work.

Community Defined Section:  Aside from Hillary's key piece, I would suggest that the book by Roland Warren -- the one that explores the notion of horizontal and vertical linkages and the loss of community -- be highlighted in this section.  In addition, I don't agree that Gene Theodori's article offers a more contemporary perspective on community.  In man respects, what Gene is proposing is in synch with what Ken Wilkinson argued in his research articles on the community.  He long advocated (as did Kaufman) that a critical attribute of community is locality oriented actions -- a so called place-base approach.  He then noted that in the absence of this, community is nonexistent or compromised.  

Engagement as a Means to Bring Forth Community: It would be helpful to have a table that list the six dimensions of the Chaskin model, a brief definition of each, and examples of each dimension. As for the PAR model, how do  you ensure that participation is broad-based and inclusive?  How do you make sure it reflects the "bridging" component that is so critical to the emergence of social capital?  While the co-creation/co-learning aspect of PAR is great, I am not seeing anything stated that argues for the need to have a diversity of voices engaged in this process.

Lines 199-201: The statement is made that many CD programs are unsuccessful in creating sustainable change. The word "many" is a loaded word.  What articles can you cite to support this statement? 

Lines 240-242: The internet presence is not simply a staff issue, is it?  Would it not also be associated with access to broadband services? 

Study Circles Approach:  I am an advocate for the Everyday Democracy approach.  In my view, action forums do produce a series of "action plans".  Is it not possible, however, that the number of issues or actions to be pursued can overwhelm a community?  That is, having too many issues to be addressed can create a certain degree of paralysis because the list is simply too large to tackle. So, the concern is how to guide a local group to pursue a reasonable list of issues that can be addressed in a timely fashion.  Not sure how well that is addressed in your communities taking part in study circles. 

Discussion: This section is solid but what I was really hoping to see was the authors connecting the MHA program to the theories discussed in the early part of the paper. To me, evidence that the social fields have slowly been developing into a community field in each site would be invaluable (or if not developing, why not?). Perhaps it is too early, but are there signs that structural shifts are beginning to take shape as represented by a broader base of people and groups that is engaged in decision-making across multiple interest areas?  The observation offered on lines 439-444 suggest that the social connections that formed prior to the MHA effort were instrumental in helping move the program forward.  To me, that is reflective of the beginning emergence of a community field.  But, I would like to see more discussion that links the results with the theoretical underpinnings of this article. As is noted on line 546, the key is the creation of long-lasting relationships, partnerships and networks.  But,  I don't see evidence offered that indicates this has occurred in the two sites, at least not yet. 

Line 466-468:  It's important to note that CES has a long history of seeking input and guidance through the use of advisory committees.  I don't see that included in the list of ways that the public has provided input to the "experts." 

Bottom line, this is a solid paper but finding a way to use the theories described in the front portion of the paper to the results from the "engagement" side of the paper, would be valuable.  Perhaps study circles helps create the opportunity for social fields to develop, but may not provide spur the emergence of a community field, at least not on a short-term basis.   

Author Response

This is a well-written article that offers some valuable points and contributions to the community development literature, with a specific focus on the Extension system in the U.S.  I do have a few items to share with the authors that I hope will further strengthen a paper that is already in decent shape.

Lines 41-42: I'm not sure I would agree that it's passive learners when it comes to the expert model.  It is more the fact the learners may not be major contributors when it comes to the content, but they are, in some cases, very active learners (such as field days). I would recommend this be modified just a bit.

This was modified

Line 43: The notion that the collaborative model is a more recent phenomenon needs to be clarified.  What is recent in terms of the time period?  I believe the CD profession (see the CD Society principles of good practice) has some sometime advocated collaboration in community work. So, "recent" does not offer much clarity as to when this happened.  I also would recommend you add citations to support this statement.

“Recent” was removed and the sentence modified to clarify.  With the other citations and discussion it was felt that a citation for the now amended paragraph was not needed.

Line 47:  I would suggest that Extension programs are not solely the result of research conducted by the LGUs.  There are several other universities and research centers that have generated knowledge that is being incorporated into the work of the LGUs outreach work.

Modified

Community Defined Section:  Aside from Hillary's key piece, I would suggest that the book by Roland Warren -- the one that explores the notion of horizontal and vertical linkages and the loss of community -- be highlighted in this section.  In addition, I don't agree that Gene Theodori's article offers a more contemporary perspective on community.  In many respects, what Gene is proposing is in synch with what Ken Wilkinson argued in his research articles on the community.  He long advocated (as did Kaufman) that a critical attribute of community is locality oriented actions -- a so called place-base approach.  He then noted that in the absence of this, community is nonexistent or compromised.

Removed the word “contemporary” and modified the paragraph.  The authors were reluctant to add more detail into an already detailed lit review so the Warren addition was not made.  What we wanted to emphasize is the place-based approach.

Engagement as a Means to Bring Forth Community: It would be helpful to have a table that list the six dimensions of the Chaskin model, a brief definition of each, and examples of each dimension. As for the PAR model, how do you ensure that participation is broad-based and inclusive?  How do you make sure it reflects the "bridging" component that is so critical to the emergence of social capital?  While the co-creation/co-learning aspect of PAR is great, I am not seeing anything stated that argues for the need to have a diversity of voices engaged in this process.

A graphic was added with the modelWe tried to emphasize the need for diversity in other sections of the article.  I am not sure there is any way you can ensure inclusive participation but you can strive to be inclusive.  More detail on this topic was perhaps moving the thought process away from the engagement focus.

Lines 199-201: The statement is made that many CD programs are unsuccessful in creating sustainable change. The word "many" is a loaded word.  What articles can you cite to support this statement?

The word was removed and the sentence rephrased.

Lines 240-242: The internet presence is not simply a staff issue, is it?  Would it not also be associated with access to broadband services?

It was a resource that the program could use to reach more people in this particular situation. 

Study Circles Approach:  I am an advocate for the Everyday Democracy approach.  In my view, action forums do produce a series of "action plans".  Is it not possible, however, that the number of issues or actions to be pursued can overwhelm a community?  That is, having too many issues to be addressed can create a certain degree of paralysis because the list is simply too large to tackle. So, the concern is how to guide a local group to pursue a reasonable list of issues that can be addressed in a timely fashion.  Not sure how well that is addressed in your communities taking part in study circles.

We did not go into a lot of detail on how the groups choose their action items because it seemed secondary to the focus of engagement and how the university can interact with community to enhance engagement.  This topic could be incorporated into another journal article perhaps?

Discussion: This section is solid but what I was really hoping to see was the authors connecting the MHA program to the theories discussed in the early part of the paper. To me, evidence that the social fields have slowly been developing into a community field in each site would be invaluable (or if not developing, why not?). Perhaps it is too early, but are there signs that structural shifts are beginning to take shape as represented by a broader base of people and groups that is engaged in decision-making across multiple interest areas?  The observation offered on lines 439-444 suggest that the social connections that formed prior to the MHA effort were instrumental in helping move the program forward.  To me, that is reflective of the beginning emergence of a community field.  But, I would like to see more discussion that links the results with the theoretical underpinnings of this article. As is noted on line 546, the key is the creation of long-lasting relationships, partnerships and networks.  But, I don't see evidence offered that indicates this has occurred in the two sites, at least not yet.

This section was enhanced with more detail and links to theories.  Specifically, a new paragraph was added to address a promising signal of long-term community field development. 

Line 466-468:  It's important to note that CES has a long history of seeking input and guidance through the use of advisory committees.  I don't see that included in the list of ways that the public has provided input to the "experts."

Advisory committees were not formed per se within the program but community coalitions or sponsoring groups were organized and helped to lead the process.  This was lifted up in the discussion section.  In the community example of Litchfield, the local Extension Committee was credited in stimulating local participation. 

Bottom line, this is a solid paper but finding a way to use the theories described in the front portion of the paper to the results from the "engagement" side of the paper, would be valuable.  Perhaps study circles helps create the opportunity for social fields to develop, but may not provide spur the emergence of a community field, at least not on a short-term basis.  

The support of an emerging community field was noted in a paragraph that was added to the discussion section.  It was linked to the community of Bradshaw and their acknowledgement of the program as the spark that is still motivating action two years following the implementation of the program.  This was acknowledged in the discussion section.

Reviewer 2 Report

See attached.

Author Response

Set 1: Article Framing and Organization, and Authors’ Positionality

This article addresses an important topic. However, I find the framing and organization to be weak, and it lacks a few key elements.

• Missing from the current draft is a clear description of what the research problem and/or questions are. What are they? This is crucial, and can’t be left out. In lines 53-57 the authors describe what they seek to do in the article, but readers are left uncertain and unclear about why and how the case example the authors focus on was chosen, and what specific questions and/or problems motivated their selection. Moreover, the abstract mentions “two examples,” and lines 55-56 mention only one example. Which is it?
• In relation to the above, the authors should reposition and expand the material in section 4 (“materials and methods”) to follow a revised introductory section that articulates the research problem and questions. This section must appear near the beginning, or readers won’t be able to make sense of article. Importantly, they also won’t be able to discern the positionality of the authors. I assume the authors were engaged in the case example(s) they studied as participants, although this is not clearly disclosed. This must be clarified.
• With respect to the results section, lines 225-227, the authors’ voice should be changed to take ownership of their agency as researchers. To write, “An illustration of an engaged model of University supported community development can be found in the MHA example” disappears the authors’ agency. The authors should write “we found” instead of “can be found”. Here, given the mysteriousness of the authors’ positionality, the lack of clarity about their research methods, and the nature of the data they generated, readers are left wondering exactly how and why they got the “results” they describe. What did you do, what “data” did you assemble and/or generate, and how? See the first bullet point above, and revise this section in close relation to your methods.

Response Bullet 1

This was strengthened in the introduction

Response Bullet 2

Done – the methods and materials section was place toward the end of the document as per the template provided by the publisher

Response Bullet 3

Done

In section 3, discussion, the authors begin by making two strong claims about what “must be” and “should not be” done that do not appear to be linked in any discernable way to their data. Where did these claims come from? Are they findings from research, or moral/political/philosophical arguments? What will happen, or not happen, if practitioners ignore what the authors claim “must be” or “should not be” done? How do we know?

In relation to what the authors wrote in their conclusion in section 5, the data the authors show us in the article isn’t enough to warrant the concluding claims they make. Moreover, since by this time I began to sense that the intended focus of the research was probably to test and/or learn from an experiment in reflective practice, it isn’t clear why they spent so much time reviewing literatures related to background concepts, without reviewing any methodological literatures related to the kind of reflective practice their study (probably) used and (perhaps) refined. My lack of clarity about all this is a sign that a major revision is required.

Response Bullet 1

The wording was modified – the strong claims are now eliminated

Response Bullet 2

These were designed as initial insights and they are now labeled as such.

Set 2: Extension History and Practice
The authors’ characterizations of extension history and practice are not accurate. They reproduce misleading assumptions that obscure many things that happen to be directly related to theme of “enhancing community.”
• The authors’ claim in lines 41-42, “Initially extension faculty primarily functioned at the expert level, sharing their knowledge with passive learners,” and other claims elsewhere in the article (see below), reflect an overly simplified story of extension that is deeply problematic and untrustworthy. It obscures the deep historical roots of two-way, reciprocal engagement in extension history that respected, drew out, and sought to strengthen the active civic agency of community members, farmers, homemakers, and youth, and it erases the many women and men who espoused and pursued it. Further, it obscures a long history of debate and disagreement about extension’s purposes and practices that continues into the present. Instead of positioning extension as having a one-way, expert dominated past which has only recently shifted to support a two-way engagement model, the authors should position it as having a long history of tension between a one-way technocratic/expert driven model and a two-way, reciprocal engagement model. Citations to support this view include the following:
o Peters, S.J. (2017). Recovering a Forgotten Lineage of Democratic Engagement. In C. Dogan, TK Eatman, and TD Mitchell (eds.), Cambridge Handbook of Service Learning and Community Engagement (New York: Cambridge University Press), pp. 71-80.
o Peters, S. J. (2014). Extension reconsidered. Choices 29:1. http://www.choicesmagazine.org/choices-magazine/theme-articles/higher-educations-roles-in-supporting-a-rural-renaissance/extension-reconsidered
o Peters, S. J. and Avila, M. (2014). Organizing for culture change through community-based research. In R. Munck, L. Mc Ilrath, B. Hall and R. Tandon (eds.), Higher Education and Community-Based Research: Creating a Global Vision (New York: Palgrave MacMillan), pp. 133-147.
o Peters, S. J. (2013). Storying and Restorying the Land-Grant System. In Roger L. Geiger and Nathan M. Sorber (Eds.), The Land-Grant Colleges and the Reshaping of American Higher Education. Perspectives on the History of Higher Education, Vol. 30. New Brunswick, NJ: Transaction Publishers, pp. 335-353.
o Peters, S. J. (2008). Reconstructing a democratic tradition of public scholarship in the land-grant system. In Brown, D., and Witte, D. (eds.) Agent of democracy: Higher education and the HEX journey. Dayton, OH: Kettering Foundation Press.
o Peters, S. J. (2006). Every farmer should be awakened: Liberty Hyde Bailey's vision of
agricultural extension work. Agricultural History, Vol. 80, No. 2: 190-219.

• What the authors write in lines 46-49 is a misleading description of extension programs, practice and experience, both historically and in contemporary times. What the authors write in these lines is an idealized depiction that rarely if ever matches actual practice and experience. It also erases tensions and disagreements between different ends and interests in extension work, and brackets out almost all of its human, social, relational, and civic/political dimensions. The authors should drastically revise their depiction of extension programs, consulting and citing the following:
o Peters, S.J., Alter, T.R., and Shaffer, T.J. (Eds.) (2018). Jumping Into Civic Life: Stories of
Public Work from Extension Professionals. Dayton, OH: Kettering Foundation Press.
o Peters, S. J. (2013). The Pursuit of Happiness, Public and Private. Preface in Ruby Green Smith, The People’s Colleges (Ithaca, NY: Cornell University Press).Peters, S.J. (2010).
Democracy and Higher Education: Traditions and Stories of Civic Engagement. East Lansing: Michigan State University Press.
o Peters, S.J., O’Connell, D., Alter, T.R., and Jack, A. (Eds.) (2006). Catalyzing Change: Profiles of Cornell Cooperative Extension Educators from Greene, Tompkins, and Erie Counties, New York. Ithaca, NY: Cornell Cooperative Extension.
o Peters, S.J., Jordan, N.R., Adamek, M. and Alter, T.R. (Eds.) (2005). Engaging Campus and Community: The Practice of Public Scholarship in the State and Land-Grant University System. Dayton, OH: Kettering Foundation Press.
o Peters, S.J. and Hittleman, M.J, (Eds.) (2003). We Grow People: Profiles of Extension
Educators, Cornell University Cooperative Extension, New York City. Ithaca, NY: Cornell Cooperative Extension.

Response Bullet 1

This section has been revised and citations added to include language that indicates the tension between the models.  The authors tried to lift up this comparison but not go into such detail that the reader is lost in the overall discussion of engagement in general.  The authors wanted to focus on  how engagement can happen at the community level to build community.  It is a judgement call on the amount of detail to include or exclude.

Response Bullet 2

This section was also revised and coordinated with the previous paragraph and bullet.  Together the authors feel this more adequately represents the evolution of involvement and approach that has been depicted in Extension systems.  There was never and intent to couch this as an “all or none” practice but rather as opportunities for variation along a continuum.  The authors feel these revisions hit a middle ground.

• What the authors write in lines 205-223 by citing Vines is misleading and untrustworthy. It’s based on a comic-book history of extension that erases disagreement, and disappears women and men who held relational, reciprocal, democratic views. See above.
• The “hybrid” model the authors write about in lines 460-465 is not new. It has long historical precedent. See above.
• What the authors write in lines 475-494 is caught up in a “program delivery” view of extension work. Such a view exists in espoused theory and practice, of course. But it is at odds with a relational organizing view that uses PAR methods and models where extension serves as facilitator and organizer rather than “conduit”, and that has been present in extension history from the late 19th century forward. It’s not new, and it’s not a recent invention. See above.

Response Bullet 1, 2 and 3

These issues are interrelated.  All three sets of text were rewritten so that the hybrid model was not listed as new but rather an evolution of delivery that has developed over time.  The tension between a more expert vs. collaborator model was indicated.   The third bullet shares the idea that engagement in Extension is not only about program delivery but also about engagement on a community organizing level even leading to and supporting democracy.  The authors acknowledge this fact but the authors purposefully tried to hone in on program delivery, an aspect that would appear to be more relevant to other academic professionals wanting to expand their level of community engagement.

Two Small Issues

The authors need to fix two small issues:
• In line 40, the citation of Peters 2002 is curious, since what the authors write has nothing to do with what that article discusses.

• The quotation that is included in lines 29-32 makes for an awkward sentence the way it is presented. Better to paraphrase to make a grammatically correct sentence, or change the punctuation to put the quotation after a colon.

Response Bullet 1

The citation was removed at this location but the body of work supports early statements and is added with other citations.

Response Bullet 2

Revised as per suggestion.

Round 2

Reviewer 2 Report

The article is improved.

Author Response

Cover letter responses to suggestions:

We hope we have addressed all of your concerns. Here are our actions...

#1 Introduction - stress goals and implications.  See pg 2 in yellow

#2 Methods and materials - add more detail on how the reflection was analyzed.  See pg 7 in yellow

#3 Heading title - changed to "Program Summary"

#4 Action Forum detail - how did it work.  See pg 8 in yellow

#5 - grammar and style.  This was initially done prior to the first revision.  It was reviewed again and changes made -- specifically, citations were given commas between authors as well as other minor changes made.  These were not highlighted in yellow, instead they were simply revised in the document.  During the course of revising some formatting or spacing could have changed.  This is very difficult to manage.  If changes were made inadvertently, we are sorry for the inconvenience.